# Modulation of hepatitis B virus pregenomic RNA stability and splicing by histone deacetylase 5 enhances viral biosynthesis

Taha Y. Taha[ID][1], Varada Anirudhan[ID][2], Umaporn Limothai[ID][3], Daniel D. Loeb[4], Pavel A. Petukhov[ID][1]*, Alan McLachlan[ID][2]*

1 Department of Pharmaceutical Sciences, College of Pharmacy, University of Illinois at Chicago, Chicago, Illinois, United States of America, 2 Department of Microbiology and Immunology, College of Medicine, University of Illinois at Chicago, Chicago, Illinois, United States of America, 3 Center of Excellence in Hepatitis and Liver Cancer, Faculty of Medicine, Chulalongkorn University, Bangkok, Thailand, 4 McArdle Laboratory for Cancer Research, University of Wisconsin - Madison, Madison, Wisconsin, United States of America

* pap4@uic.edu (PAP); mclach@uic.edu (AM)

**Data Availability Statement:** All relevant data are within the manuscript and its Supporting Information files.

## Abstract

Hepatitis B virus (HBV) is a worldwide health problem without curative treatments. Investigation of the regulation of HBV biosynthesis by class I and II histone deacetylases (HDACs) demonstrated that catalytically active HDAC5 upregulates HBV biosynthesis. HDAC5 expression increased both the stability and splicing of the HBV 3.5 kb RNA without altering the translational efficiency of the viral pregenomic or spliced 2.2 kb RNAs. Together, these observations point to a broader role of HDAC5 in regulating RNA splicing and transcript stability while specifically identifying a potentially novel approach toward antiviral HBV therapeutic development.

## Author summary

This study demonstrates that HDAC5 deacetylation of host cellular factor(s) results in increased HBV biosynthesis by enhancing viral transcript stability and splicing via direct or indirect binding of host factors to viral intron sequences. This represents the first demonstration of this type of post-transcriptional regulation in the liver and is similar to observations seen for cellular transcripts in neural and cardiac cell types. These observations suggest a more general phenomenon which could represent an additional post-transcriptional code governing the regulation of RNA:protein interactions and hence RNA metabolism. Therefore, covalent modifications of RNA binding proteins may modulate post-transcriptional gene expression in an analogous manner to the known histone code that controls gene transcription. Although this analysis primarily relates to the mechanism(s) by which HDAC5 governs HBV RNA metabolism, it does have significant therapeutic implications. The inhibition of HDAC5 in combination with current nucleos(t)ide analog drugs targeting the viral reverse transcriptase/DNA polymerase might aid in the

**Funding:** This work was supported by the National Institutes of Health grants R01 AI125401 (AMcL), R01 CA238328 (AMcL), and R01 HL130760 (PAP), the PhRMA Foundation Fellowship for Pharmacology and Toxicology (TYT), and the Thailand Research Fund (TRF) Senior Research Scholar RTA6280004 (UL). We also are indebted to Dr. Greg Thatcher and the UICentre for Drug Discovery for financial support. The funders had no role in study design, data collection and analysis, decision to publish, or preparation of the manuscript.

**Competing interests:** The authors have declared that no competing interests exist.

treatment and possible resolution of chronic infections by targeting both host and viral factors.

## Introduction

HBV infection is a worldwide health problem and is endemic in many regions of Asia and Africa [1,2]. Infection can be prevented by vaccination (90–95% efficacy), but chronic infection remains a major clinical problem with 200 to 500 million cases worldwide and 1 million deaths annually [2]. To date, there is no reliable curative therapy [1,3]. One of the critical barriers to the development of novel HBV therapeutics is the lack of targetable host factors that are necessary for infection and viral propagation.

HBV is a small enveloped hepadnavirus that replicates via reverse transcription of the pregenomic 3.5 kb RNA within its capsid to generate the viral 3.2 kbp partially double-stranded DNA genome [4,5]. The major viral transcripts reported in natural infection in humans, transgenic mice, and cell culture are the HBV 3.5 and 2.1 kb RNAs [6–8]. The viral 3.5 kb RNA represents two distinct transcripts differing by approximately 30 nucleotides at the 5'-end of the RNA. The precore 3.5 kb RNA encodes the HBV early antigen (HBeAg) and the pregenomic 3.5 kb RNA is translated to produce the HBV core antigen (HBcAg) and the viral polymerase [6–8]. The pregenomic RNA also serves as the template for viral reverse transcription in the initial steps of replication [4,5]. The 2.1 kb RNA codes for the middle and major HBV surface antigens (HBsAg) [6–8]. Two minor transcripts of 2.4 and 0.7 kb encode the large HBsAg [6–8] and the HBV X gene polypeptide [9,10]. In addition, a variety of spliced viral RNAs of undefined importance in the HBV life cycle have been reported [6,11–15]. A major singly spliced HBV 2.2 kb RNA encodes a truncated HBcAg lacking the terminal cysteine residue [6,11–13,16]. In addition, a novel open reading frame within this spliced RNA may be translated to generate a polypeptide (HBSP) comprising the first 46 amino acids of the viral polymerase and an additional 47 carboxyl-terminal amino acids [17]. This protein does not appear to be essential for either viral transcription and replication [17].

Histone deacetylases (HDACs) are a family of 18 enzymes that catalyze the removal of acetyl groups from histones and non-histone proteins [18,19]. HDACs are classified based on sequence homology to yeast HDACs and are divided into classical zinc-dependent HDACs and NAD$^+$-dependent sirtuins (SIRTs) [20]. Classical HDACs are further subdivided into class I (HDACs 1–3 and 8), class IIa (HDACs 4, 5, 7, and 9), class IIb (HDACs 6 and 10), and class IV (HDAC 11) HDACs. Class I HDACs are generally regarded as the main deacetylases of histones among other roles in the cell, while class IIa and IIb HDACs are involved in a variety of cellular processes including cell cycle, DNA repair, differentiation, and apoptosis in deacetylase enzyme activity-dependent and -independent mechanisms [21–26]. There have been multiple reports linking HDACs to the accessibility of the transcriptional machinery to the HBV covalently closed circular DNA (cccDNA) present in cells as a nuclear chromatin template [27–32]. Interestingly, single-cell analysis of hepatocytes [33] revealed that the expression of HDAC5 correlates closely with the distinct zonal profile of viral biosynthesis [34] across the liver lobule in HBV transgenic mice (S1 Fig). The levels of the other classical HDAC transcripts are considerably lower than HDAC5 and they are not differentially expressed across the liver lobule (S1B Fig). Furthermore, there is no reported effect of class II HDACs on HBV biosynthesis.

In this study, the effect of class I, IIa, and IIb HDACs on HBV biosynthesis is evaluated. It is demonstrated that HDAC5 significantly upregulates HBV biosynthesis by increasing the

stability and splicing of the HBV 3.5 kb RNA in a catalytic activity dependent manner. These observations indicate that HDAC5 is a potential host factor supporting viral biosynthesis and HDAC5 selective inhibitors might be developed as antiviral therapeutics.

## Results

### HDAC5 increases HBV biosynthesis in the human hepatoma HepG2 and nonhepatoma HEK293T cell lines

Previous studies had suggested a role for HDAC1 in the repression of HBV transcription, but the effects of other class I and class II HDACs on viral biosynthesis have not been considered. HepG2 cells support HBV transcription and replication when transfected with the HBV DNA (4.1 kbp) construct (Fig 1A). To study the impact of HDACs on HBV biosynthesis, constructs expressing representatives of class I (HDACs 1, 3, and 8), class IIa (HDACs 4, 5, and 7), and class IIb (HDAC6) HDACs were utilized. The effects of HDACs 1 and 3–8 expression on viral 3.5 kb RNA, HBV DNA replication intermediates, HBV core polypeptide (p21) and capsid levels were modest and generally less than 2-fold (Fig 1A). However, HDAC5 increased capsid abundance by 3-fold and HDAC7 significantly decreased viral RNA, core polypeptide (p21), and capsid levels (Fig 1A). As the effects of HDACs 1 and 3–8 expression in HepG2 cells were modest (Fig 1A), it is challenging to ascertain the potential effect of individual HDACs on HBV biosynthesis in this system.

In contrast to HepG2 cells, transfection of HEK293T cells with the HBV DNA (4.1 kbp) construct resulted in detectable levels of HBV RNA but barely detectable viral DNA replication intermediates (Fig 1B). When HEK293T cells are transfected with HDACs 1 and 3–8 expression vectors, HBV 3.5 kb RNA levels did not change greatly in the presence of HDACs 1, 3, 5, 6, and 8, but decreased approximately 2- and 4-fold with HDACs 4 and 7, respectively (Fig 1B). Interestingly, the HBV 2.1 kb RNA levels appeared to increase approximately 1.5-fold with HDAC5 expression (Fig 1B). The viral DNA replication intermediate levels increased 3- and 6-fold with HDAC1 and 5, respectively, but was absent or marginally detectible for the other HDACs (Fig 1B). The HBV core polypeptide (p21) levels increased 5-fold with HDAC5 expression but decreased 3- to 6-fold with the expression of other HDACs. HDAC1 failed to modulate core polypeptide (p21) levels to a major extent while supporting an approximately 4-fold increase in capsid formation (Fig 1B). However, HBV capsids were increased approximately 50-fold with HDAC5 expression but were undetectable in the presence of the other HDACs (Fig 1B). The observation that an HDAC5-mediated 5-fold increase in HBV core polypeptide (p21) levels is associated with a 50-fold enhancement in HBV capsid abundance is consistent with previous reports indicating that the conversion of HBcAg dimers into capsids occurs by a highly cooperative process [35–37]. Moreover, the limited expression of HDAC5 protein in HepG2 cells compared with HEK293T cells (S1C Fig) may explain the modest effect of HDAC5 expression on HBV biosynthesis in HepG2 cells compared with the robust effect in HEK293T cells.

### HDAC5 catalytic activity upregulates HBV biosynthesis

Given their relatively low enzymatic activities, class IIa HDACs are known for their enzymatic activity-independent biological roles, such as scaffolding factors within corepressor complexes [22–26]. To evaluate the role of HDAC5 catalytic activity in the enhancement of HBV biosynthesis in HEK293T cells, LMK235, a class IIa selective HDAC inhibitor [38], was evaluated for its effect on HBV biosynthesis. HDAC5 had little effect, less than 2-fold, on the level of the viral 3.5 kb RNA, but dramatically increased the level of viral replication intermediates (Fig 2A, lanes

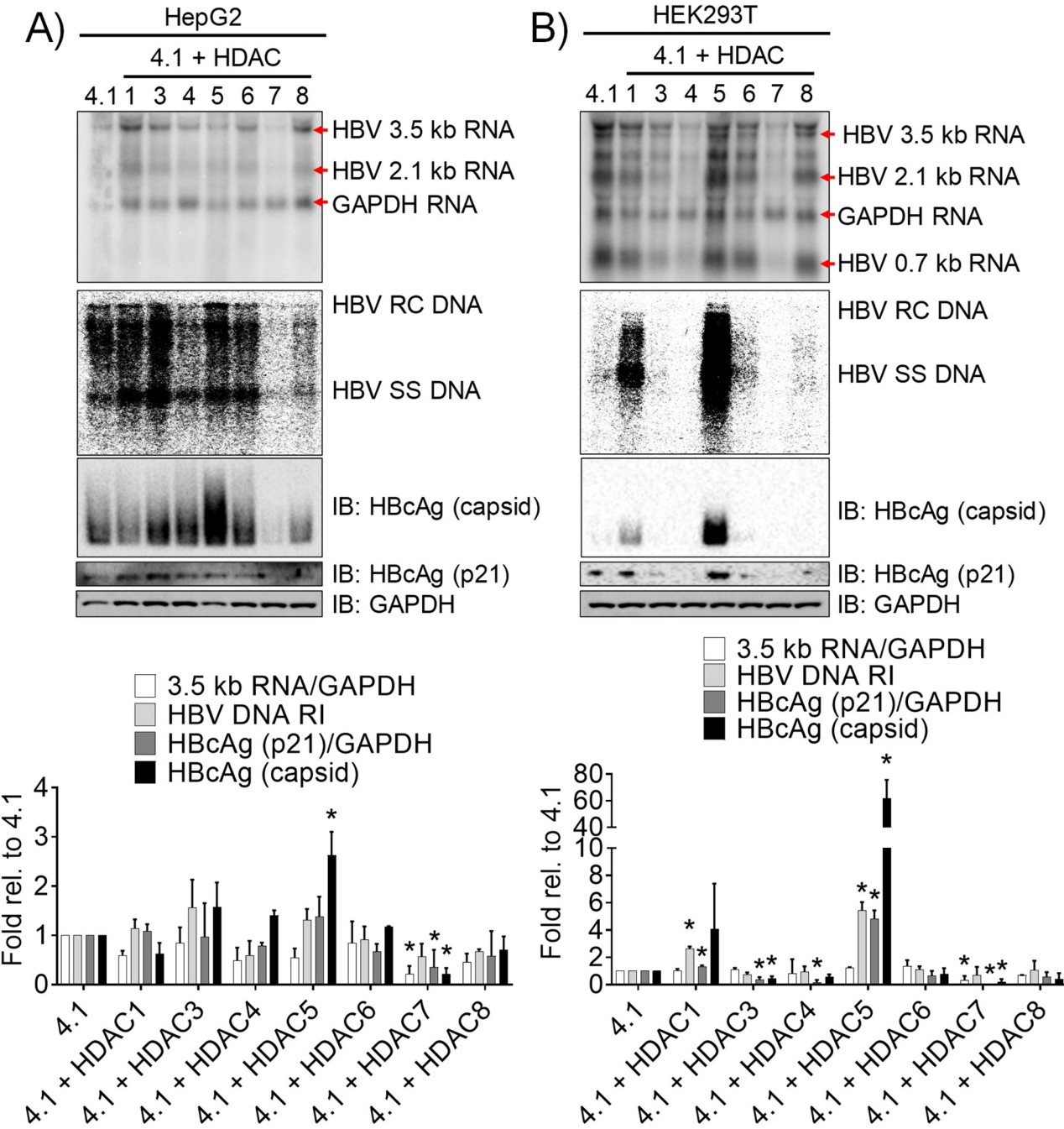

**Fig 1. HDAC5 increases HBV biosynthesis in cell culture.** (A) HepG2 and (B) HEK293T cells were transfected with the HBV DNA (4.1 kbp) construct with or without HDAC expression vectors. Total cellular RNA, viral DNA replication intermediates, and cytoplasmic proteins were analyzed by RNA (Northern) filter hybridization, DNA (Southern) filter hybridization, and Western blot analyses, respectively. The GAPDH transcript was used as an internal control for RNA loading per lane. The 3.9 kb transcript observed above the HBV 3.5 kb RNA probably represents the previously reported HBV long xRNA that initiates from the X promoter region [79]. HBV RC DNA, HBV relaxed circular DNA; HBV SS DNA, HBV single-stranded DNA. Quantitation of the HBV 3.5 kb RNA relative to the GAPDH RNA, the HBV DNA replication intermediates, the HBV native capsids, and the core polypeptide (p21) levels relative to GAPDH is shown as mean plus standard deviation from two independent analyses. Levels that are statistically significantly different than the levels in cells transfected with the HBV DNA (4.1 kbp) construct only, as determined by Student's t test (P < 0.05), are indicated with an asterisk.

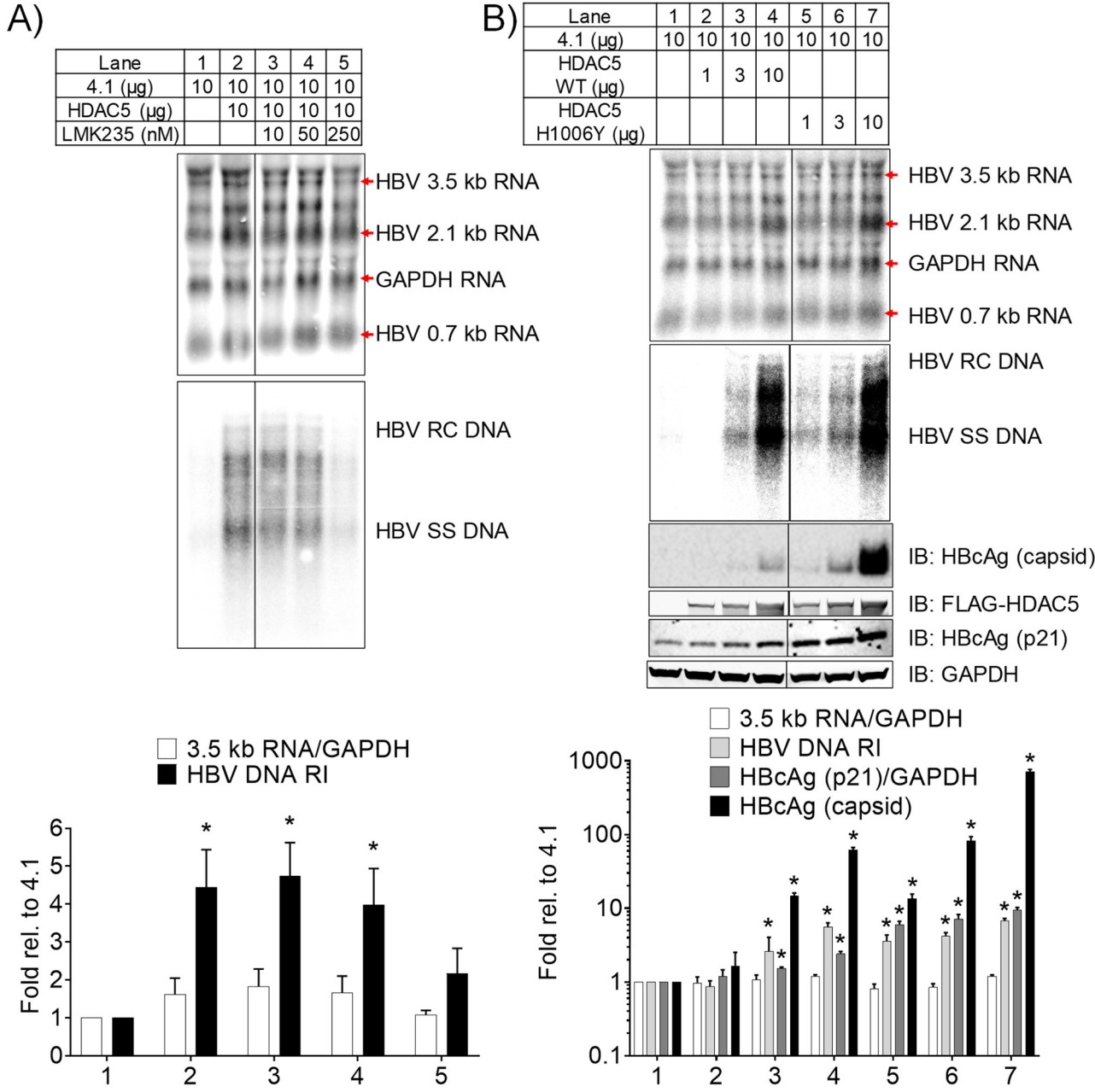

**Fig 2. HDAC5 enzymatic activity is required for the upregulation of HBV biosynthesis.** (A) HEK293T cells were transfected with the HBV DNA (4.1 kbp) construct with or without the HDAC5 expression vector and LMK 235 treatment. (B) HEK293T cells were transfected with the HBV DNA (4.1 kbp) construct with or without the HDAC5 expression vectors encoding the wild type or enhanced activity mutant HDAC5 (H1006Y) proteins. The black lines indicate noncontiguous lanes from a single blot. Total cellular RNA, viral DNA replication intermediates, and cytoplasmic proteins were analyzed by RNA (Northern) filter hybridization, DNA (Southern) filter hybridization, and Western blot analyses, respectively. The GAPDH transcript was used as an internal control for RNA loading per lane. The 3.9 kb transcript observed above the HBV 3.5 kb RNA probably represents the previously reported HBV long xRNA that initiates from the X promoter region [79]. HBV RC DNA, HBV relaxed circular DNA; HBV SS DNA, HBV single-stranded DNA. Quantitation of the HBV 3.5 kb RNA relative to GAPDH RNA, the HBV DNA replication intermediates, the HBV native capsids, and the core polypeptide (p21) levels relative to GAPDH is shown as mean plus standard deviation from two independent analyses. Levels that are statistically significantly different than the levels in cells transfected with the HBV DNA (4.1 kbp) construct only, as determined by Student's t test ($P < 0.05$), are indicated with an asterisk.

1 and 2). Inhibition of HDAC5 activity by LMK235 led to a concentration-dependent decrease in HBV DNA replication intermediate levels. Indeed, viral replication intermediates were decreased dramatically to levels similar to those observed in cells transfected with the HBV DNA (4.1 kbp) construct alone (Fig 2A). The levels of viral 3.5 kb RNA changed less than 2-fold when the cells were treated with LMK235 (Fig 2A). The level of the HBV 2.1 kb RNA increased modestly with HDAC5 expression whereas LMK235 treatment reversed this effect (Fig 2A).

To investigate further the requirement of HDAC5 catalytic activity for HBV biosynthesis, an HDAC5 expression vector carrying a histidine-to-tyrosine mutation at position 1006 (H1006Y), which has been shown to increase the catalytic activity of HDAC5 more than 50-fold, was utilized [39]. The wild-type HDAC5 did not alter the level of viral 3.5 kb RNA. In contrast, the level of viral core polypeptide (p21), capsids, and replication intermediates were increased when 3 or 10 μg of the wild-type HDAC5 expression vector was utilized (Fig 2B). However, when the mutant HDAC5 (H1006Y) expression vector was used, 1 μg of the expression vector supported viral biosynthesis. Additional increases in this expression vector led to further increases in HBV core polypeptide (p21) and capsid synthesis compared with the wild type HDAC5 expression vector (Fig 2B). Of note, when 10 μg of the HDAC5 (H1006Y), and to a lesser extent 10 μg of the wild-type HDAC5 expression vector, are cotransfected with HBV DNA (4.1 kbp) construct, there was an apparent 2-fold increase in HBV 2.1 kb RNA (Fig 2B). These observations indicate that the catalytic function of HDAC5 contributes to enhanced HBV biosynthesis.

## HDAC5 increases the abundance of HBcAg encoding HBV RNAs

The modest HDAC5-mediated increase in total HBV RNAs (Figs 1 and 2) and specifically the apparent increase in the viral 2.1 kb RNA suggests a possible RNA-mediated enhancement of viral biosynthesis. It is unclear if the apparent increase in the viral 2.1 kb RNA band represents an increase in the HBV 2.1 kb transcript encoding the middle and major HBsAg or if it is due to the appearance of the singly spliced HBV 2.2 kb RNA derived from the HBV 3.5 kb RNA [6,11–13,16]. Indeed, mutation of each of the viral open reading frames alone or in various combinations except for the core polypeptide coding region in the context of the replication competent pCMVHBV and HBV DNA (4.1 kbp) constructs fails to inhibit the HDAC5-mediated increase in HBV capsid abundance in HepG2 cells (S2 Fig). Moreover, HDAC5 expression failed to increase capsid production from core polypeptide (p21) sub-genomic expression vectors (Fig 3A and 3B). These observations suggest that HBV RNA splicing might contribute to the HDAC5-mediated enhancement of HBV biosynthesis through the post-transcriptional regulation of viral transcript splicing and/or stability. To investigate further this possibility, a CMV-driven HBV construct, pCMVHBV, was used to obtain robust levels of HBV 3.5 kb RNA synthesis in the presence of more limited levels of other viral RNAs [40]. This approach permitted a more accurate investigation of HBV 3.5 and spliced 2.2 kb transcripts by RNA (Northern) filter hybridization analysis. Transfection of HEK293T cells with the pCMVHBV construct produced a prominent HBV 3.5 kb RNA band with much lower abundance of other viral transcripts (Fig 3C). In the presence of HDAC5 expression, the abundance of the viral 3.5 kb RNA changed by less than 2-fold, but an HBV 2.2 kb RNA was increased approximately 7-fold (Fig 3C). In addition, HDAC5 expression led to a 3-fold increase in HBV capsid levels (Fig 3C). To determine the level of viral RNAs initiating from the HBV pregenomic RNA transcription start site, primer extension analysis using a $^{32}$P-labeled DNA probe spanning HBVayw nucleotide coordinates 1976–1941 was performed. Expression of HDAC5 in the presence of the pCMVHBV construct in HEK293T cells led to a 2- to 6-fold increase in HBV transcript initiating from the HBV pregenomic RNA start site (Fig 3D). Similarly, cotransfection of

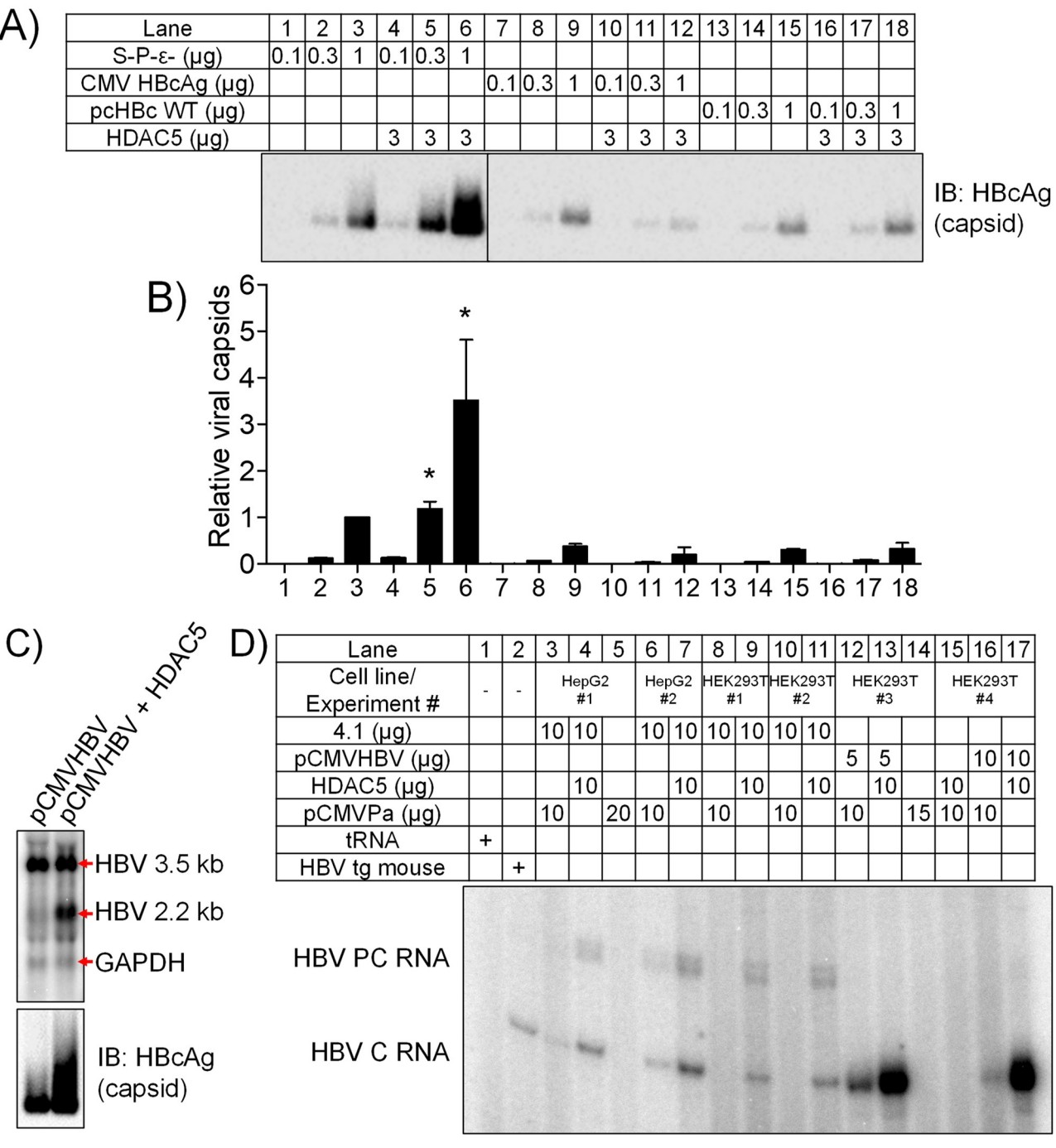

**Fig 3. HDAC5 increases the abundance of HBcAg-encoding HBV RNAs.** (A) HepG2 cells were transfected with a pCMVHBV construct that cannot synthesize surface and polymerase polypeptides and does not contain the encapsidation signal ε (S-P-ε-), or sub-genomic CMV-driven HBV core protein only expression vectors with or without the HDAC5 expression vector. The viral proteins present in the cytoplasmic extracts were analyzed by Western blot analysis for native viral capsids. The black lines indicate noncontiguous lanes from a single blot. (B) Quantitation of viral capsids relative to lane 3 which is designated as a relative abundance of 1.0. Mean plus standard deviation for 2 independent experiments is shown. Comparison of the levels of the viral capsids in the presence or absence of HDAC5 expression that are statistically significantly different, as determined by a paired Student's t test (P < 0.05), are indicated with an asterisk. (C) HEK293T cells were transfected with the pCMVHBV expression vector with or without the HDAC5 expression vector. Total cellular RNA and protein were collected and analyzed by RNA (Northern) filter hybridization and Western blotting analyses, respectively. The GAPDH transcript was used as an internal control for RNA loading per lane. (D) RNA from HepG2 or HEK293T cells transfected with the HBV DNA (4.1 kbp) or pCMVHBV constructs with or without the HDAC5 expression vector as indicated was analyzed by primer extension. HBV PC RNA, HBV precore RNA; HBV C RNA, HBV core (pregenomic) RNA. 10 μg tRNA was included as a negative control and 10 μg of HBV transgenic (tg) mouse liver RNA was included as a positive control.

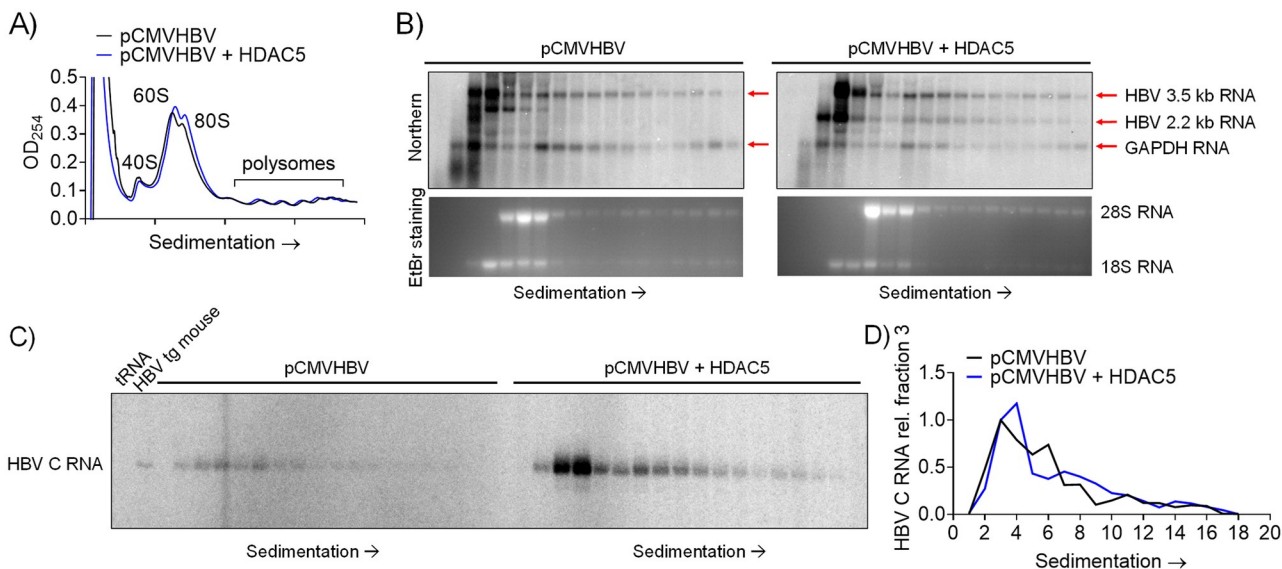

**Fig 4. HDAC5 does not alter the translational efficiency of HBV RNAs.** HEK293T cells were transfected with the pCMVHBV construct with or without the HDAC5 expression vector. Polysomes were subjected to sucrose density ultracentrifugation. (A) Polysome profile measured by absorption at 254 nm. (B) RNA isolated from gradient fractions was analyzed by RNA (Northern) filter hybridization analysis. Ethidium bromide (EtBr) staining indicates the positions of the 28S and 18S ribosomal RNAs. Red arrows indicate the position of HBV 3.5 and 2.2 kb RNAs and GAPDH RNA. The band below the 3.5 kb band in the pCMVHBV sample is potentially contaminant plasmid DNA. (C) RNA isolated from gradient fractions was analyzed by primer extension. 10 μg tRNA was included as a negative control and 10 μg of HBV transgenic (tg) mouse liver RNA was included as a positive control. HBV C RNA, HBV core (pregenomic) RNA. (D) Quantitation of the primer extension analysis. Abundances of the primer extension products are estimated relative to the product in fraction 3 which is designated as a relative abundance of 1.0.

the HBV DNA (4.1 kbp) construct with the HDAC5 expression vector in HepG2 or HEK293T cells led to increase in transcription initiation from the HBV pregenomic and precore RNA start sites (Fig 3D). These data suggest that HDAC5 upregulates the abundance of HBV RNAs initiating from the HBV pregenomic RNA transcription start site including both the HBV 3.5 kb and spliced 2.2 kb RNAs that encode for the HBV core protein [6,11–13,16].

## HDAC5 does not alter the translational efficiency of HBV RNAs

To ascertain whether the putative HBV spliced 2.2 kb RNA is translated and if HDAC5 modulates viral RNA translational efficiency, polysome profiling analysis was conducted in HEK293T cells transfected with the pCMVHBV construct. HDAC5 expression did not alter the overall polysome profile (Fig 4A). The HBV 3.5 kb RNA was distributed across the complete spectrum of polysomes in an HDAC5-independent manner suggesting efficient translation (Fig 4B). In the absence of HDAC5 expression, the HBV spliced 2.2 kb RNA was not detected in polysomes and was marginally detectible in the 80S monosome fraction (Fig 4B). In the presence of HDAC5 expression, the HBV spliced 2.2 kb RNA was readily detectible in polysomes, displaying a similar distribution to the viral 3.5 kb RNA (Fig 4B). To support the hypothesis that the HBV spliced 2.2 kb RNA maps to the same initiation site as the HBV pregenomic RNA, the polysome profile fractions were subjected to primer extension analysis. Overall, the expression of HDAC5 led to an approximately 5-fold increase in abundance of the HBV 3.5 and 2.2 kb RNAs, while the distribution of these transcripts was unaffected across the spectrum of polysomes. These data suggest that HDAC5 increases the abundance of the HBV 3.5 and 2.2 kb transcripts, without affecting their translational efficiency (Fig 4).

## HDAC5 increases the abundance of the HBV spliced 2.2 kb RNA

To confirm that the HBV spliced 2.2 kb RNA observed in the RNA (Northern) filter hybridization analysis and polysome profiles is the previously described HBV spliced 2.2 kb RNA [6,11–13,16], a semi-quantitative PCR analysis using primers spanning the splice sites and generating a 163 bp product from this spliced viral RNA was performed. No DNA product of the correct size was detected when HEK293T cells were transfected with the HBV DNA (4.1 kbp) construct (Fig 5A). In contrast, the expected PCR product was observed when HDAC5 was expressed with the HBV DNA (4.1 kbp) construct (Fig 5A). In addition, the expected PCR product was detected irrespective of HDAC5 expression with the pCMVHBV construct (Fig 5A). To quantify the effect of HDAC5 on the abundance of the HBV 3.5 and spliced 2.2 kb RNAs derived from the pCMVHBV construct, RT-qPCR analysis of RNA isolated from HEK293T cells transfected with the pCMVHBV construct with or without the HDAC5 expression vector was performed. PCR primers specific for the HBV pregenomic 3.5 kb plus spliced 2.2 kb RNAs (HBVayw nucleotide coordinates 2311–2420), the unspliced HBV 3.5 kb RNA but not the HBV spliced 2.2 kb RNA (HBVayw nucleotide coordinates 2590–2772), and the HBV spliced 2.2 kb RNA but not the HBV 3.5 kb RNA (HBVayw nucleotide coordinates

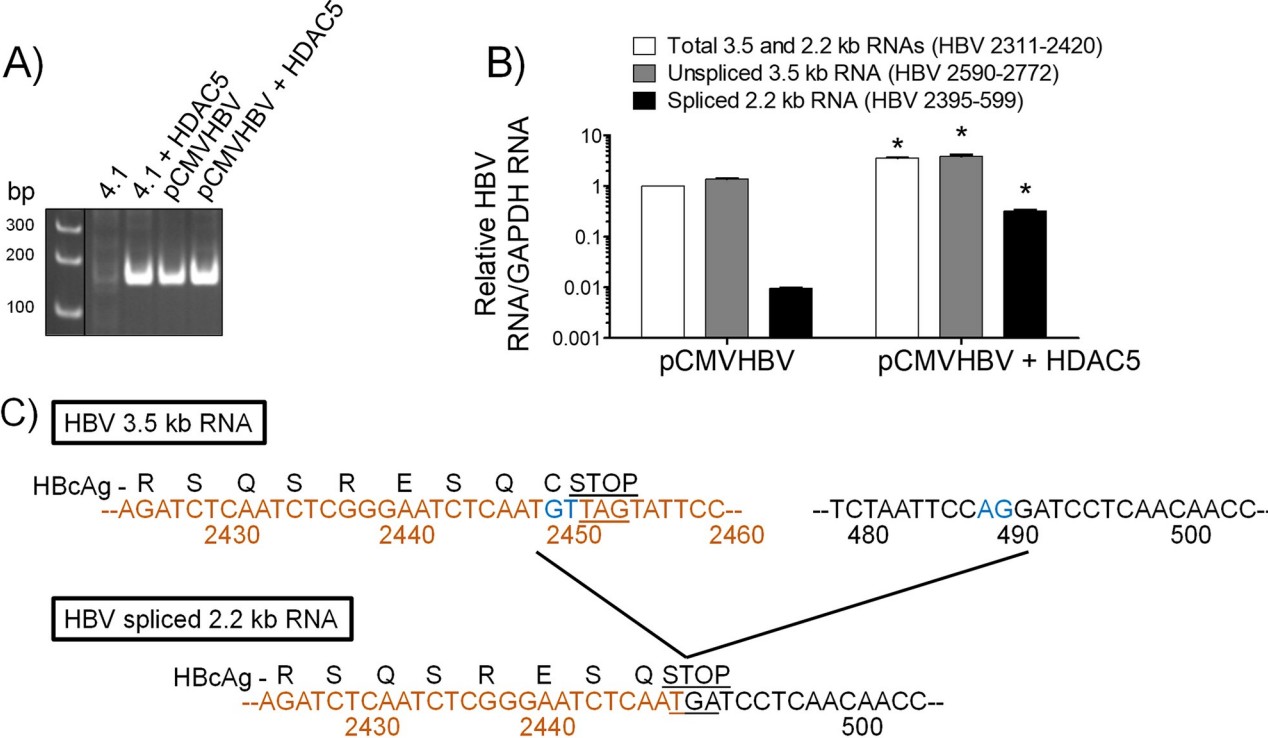

**Fig 5. HDAC5 increases the abundance of both the HBV spliced 2.2 kb and HBV 3.5 kb RNAs.** (A) HEK293T cells were transfected with the HBV DNA (4.1 kbp) or pCMVHBV constructs with or without the HDAC5 expression vector. PCR analysis of total RNA was conducted with forward primer spanning HBVayw coordinates 2395–2414 and reverse primer spanning HBVayw coordinates 599–580 [80]. Amplification products were resolved on a 6% polyacrylamide gel. The black lines indicate noncontiguous lanes from a single gel. (B) RT-qPCR analysis of RNA isolated from HEK293T cells transfected with the pCMVHBV construct with or without the HDAC5 expression vector with primers spanning the indicated HBVayw sequences [80]. The signal normalized to the GAPDH control is shown as mean plus standard deviation from two replicate analyses. Pair-wise comparison of the levels of the PCR products in the presence or absence of HDAC5 expression that are statistically significantly different in cells transfected with the pCMVHBV construct, as determined by Student's t test (P < 0.05), are indicated with an asterisk. (C) The 163 bp PCR product shown in A was sequenced. The HBV DNA sequence encoding the 5' and 3' RNA sequences are shown in orange and black, respectively. The characteristic GT splice donor and AG splice acceptor sites are shown in blue. By splicing the HBV 3.5 kb transcript into the 2.2 kb transcript, the terminal cysteine codon of the core polypeptide sequence is replaced with a newly created TGA translation stop codon.

2395–599) were utilized. HDAC5 led to an approximately 3-fold increase in the levels of HBV 3.5 and 2.2 kb RNAs and unspliced 3.5 kb RNA, whereas the HBV spliced 2.2 kb RNA abundance was increased approximately 33-fold (Fig 5B). To validate further that the PCR product was derived from the HBV spliced 2.2 kb RNA, this PCR product was sequenced [41]. The sequencing analysis demonstrated that the PCR product was derived from the HBV spliced 2.2 kb RNA, which encodes an HBV core polypeptide lacking the terminal cysteine residue (Fig 5C). This truncated HBV core polypeptide appears to support capsid assembly and viral replication slightly more efficiently than the wild type HBV core polypeptide (Fig 6). Additionally, modulating the ratio of the wild type to truncated HBV core polypeptide appears to increase capsid assembly to a modest extent without affecting viral replication (Fig 6). These data suggest that HDAC5 simultaneously increased the abundance of both the HBV 3.5 kb pregenomic RNA and the spliced 2.2 kb RNA, which codes for a functional HBV core polypeptide.

## HDAC5 increases the stability of HBV RNAs

To determine if the greater abundance of HBV 3.5 kb and spliced 2.2 kb RNAs associated with HDAC5 expression is due to increased synthesis, the transcription from the HBV core promoter was measured by HBV core promoter-driven luciferase constructs. HDAC5 did not affect the transcriptional activity of the HBV core promoter (S3 Fig). To ascertain whether the stability of the HBV 3.5 kb and spliced 2.2 kb RNAs stability is affected by HDAC5 expression, HBV RNA turnover was measured after the inhibition of transcription by actinomycin D (Fig 7A). HDAC5 expression stabilized the HBV 3.5 kb and spliced 2.2 kb RNAs. The half-lives of the HBV 3.5 kb and spliced 2.2 kb RNAs were improved approximately 2- and 3-fold, respectively, when HDAC5 was expressed (Fig 7B). To examine the HBV 3.5 kb RNA independently of the HBV spliced 2.2 kb RNA, a probe spanning HBVayw nucleotide coordinates 2451–492 was utilized (Fig 7A, bottom panel). As expected, this probe detected the HBV 3.5 kb RNA but not the HBV spliced 2.2 kb RNA (Fig 7A). Quantification of the HBV 3.5 kb RNA half-life using this sub-genomic probe led to similar half-life estimation to that obtained using the complete HBV genome as probe (Fig 7). Overall these data demonstrate that HDAC5 improves both the stability and splicing of the HBV 3.5 kb RNA, thereby increasing the abundance of the wild-type and mutant (lacking terminal-cysteine residue) core polypeptides, which support greater viral replication (Fig 6; [42]).

## Discussion

HDACs have been implicated as regulators of alternative splicing, although a clear mechanism has not been established [43]. HDAC5, in particular, has been shown to be associated with alternative splicing of *Bdnf* transcripts in neuronal tissue [44] and several cardiomyocyte gene transcripts in cardiac tissue [45]. However, a role for HDAC5, the highest expressed classical HDAC in hepatocytes (S1 Fig), in RNA processing in the liver has not been reported. In this study, HDAC5 uniquely supported robust HBV biosynthesis in a deacetylase-dependent manner and was amenable to inhibition by a small molecule inhibitor (Figs 1 and 2). HDAC5 expression led to elevated levels of the HBV spliced 2.2 kb RNA that encodes a core protein lacking the terminal cysteine residue (Fig 3C). Although upregulation of this transcript may explain greater core polypeptide (p21) and capsid levels, it does not explain the increase in HBV replication as this transcript does not encode a functional viral reverse transcriptase/ DNA polymerase and cannot be reverse transcribed to generate relaxed circular HBV 3.2 kb DNA [13]. Analysis of the levels of correctly initiated HBV pregenomic RNA by primer extension in combination with RT-qPCR analysis demonstrated that the abundance of the HBV 3.5 kb RNA was also increased (Figs 3D and 5B). Furthermore, both transcripts were efficiently

| Lane | 1 | 2 | 3 | 4 | 5 | 6 | 7 | 8 | 9 | 10 | 11 | 12 | 13 | 14 | 15 |
|---|---|---|---|---|---|---|---|---|---|---|---|---|---|---|---|
| C- (µg) | 1 | 1 | 1 | 1 | 1 | 1 | 1 | 1 | 1 | 1 | 1 | 1 | 1 | 1 | 1 |
| S-P-ε- (µg) | 0.1 | 0.3 | 1 | | | | | | | 1 | 1 | 0.3 | 0.3 | 0.3 | 0.3 |
| S-P-ε-ΔC183 (µg) | | | | 0.1 | 0.3 | 1 | | | | 0.3 | | 0.3 | | 1 | |
| S-P-ε-ΔC183(2.2) (µg) | | | | | | | 0.1 | 0.3 | 1 | | 0.3 | | 0.3 | | 1 |

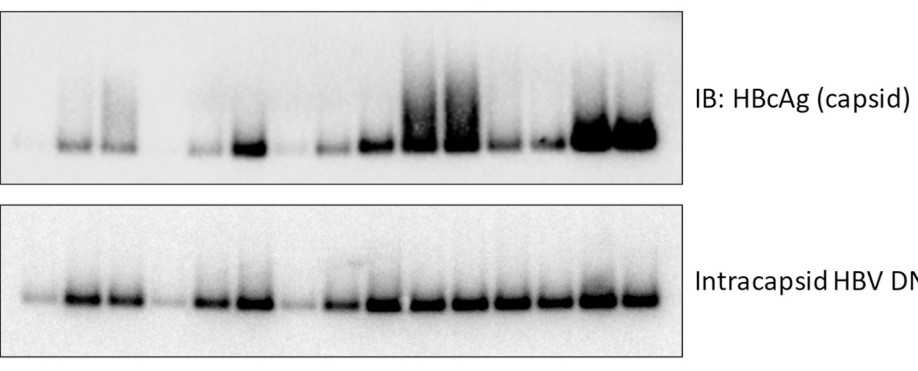

IB: HBcAg (capsid)

Intracapsid HBV DNA RI

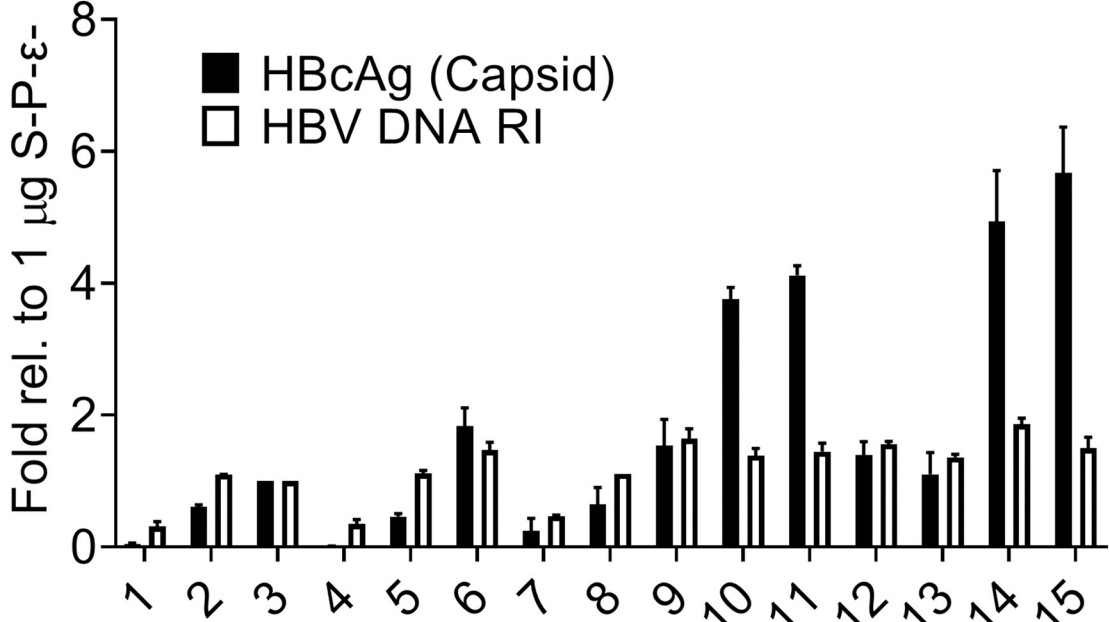

**Fig 6. HBV core particles lacking C183 are replication competent.** HEK293T cells transfected with the pCMVHBV construct lacking expression of the core polypeptide (C-) and mutant pCMVHBV constructs that lack expression of the surface and polymerase polypeptides and the encapsidation signal ε (S-P-ε-) in the presence or absence of the HDAC5 expression vector. Construct S-P-ε- expresses both the wild-type core and core polypeptide lacking the terminal cysteine residue from the 3.5 and spliced 2.2 kb HBV RNAs, respectively. Constructs S-P-ε-ΔC183 (possessing a mutation of the splice donor site plus a premature termination codon) and S-P-ε-ΔC183(2.2) (lacking the intron sequence) express the core polypeptide lacking the terminal cysteine residue from the 3.5 or 2.2 kb HBV RNAs, respectively. The viral capsid and intracapsid HBV DNA replication intermediates were analyzed from cytoplasmic extracts by Western blot and DNA (Southern) filter hybridization analyses, respectively. Quantitation of HBcAg (capsids) and HBV DNA replication intermediates are shown as means plus standard deviations from two independent analyses.

translated as ascertained by polysome profiling analysis suggesting translational regulation was not a major factor contributing to enhanced HBV core polypeptide (p21) and capsid levels (Fig 4). It is generally considered that one of the major non-coding functions of RNA splicing is to increase RNA stability although the precise mechanisms may depend on a variety of distinct pathways [46]. Inhibition of transcription by actinomycin D indicated that HDAC5 expression increased the half-life of the HBV 3.5 and spliced 2.2 kb RNAs (Fig 7). Together,

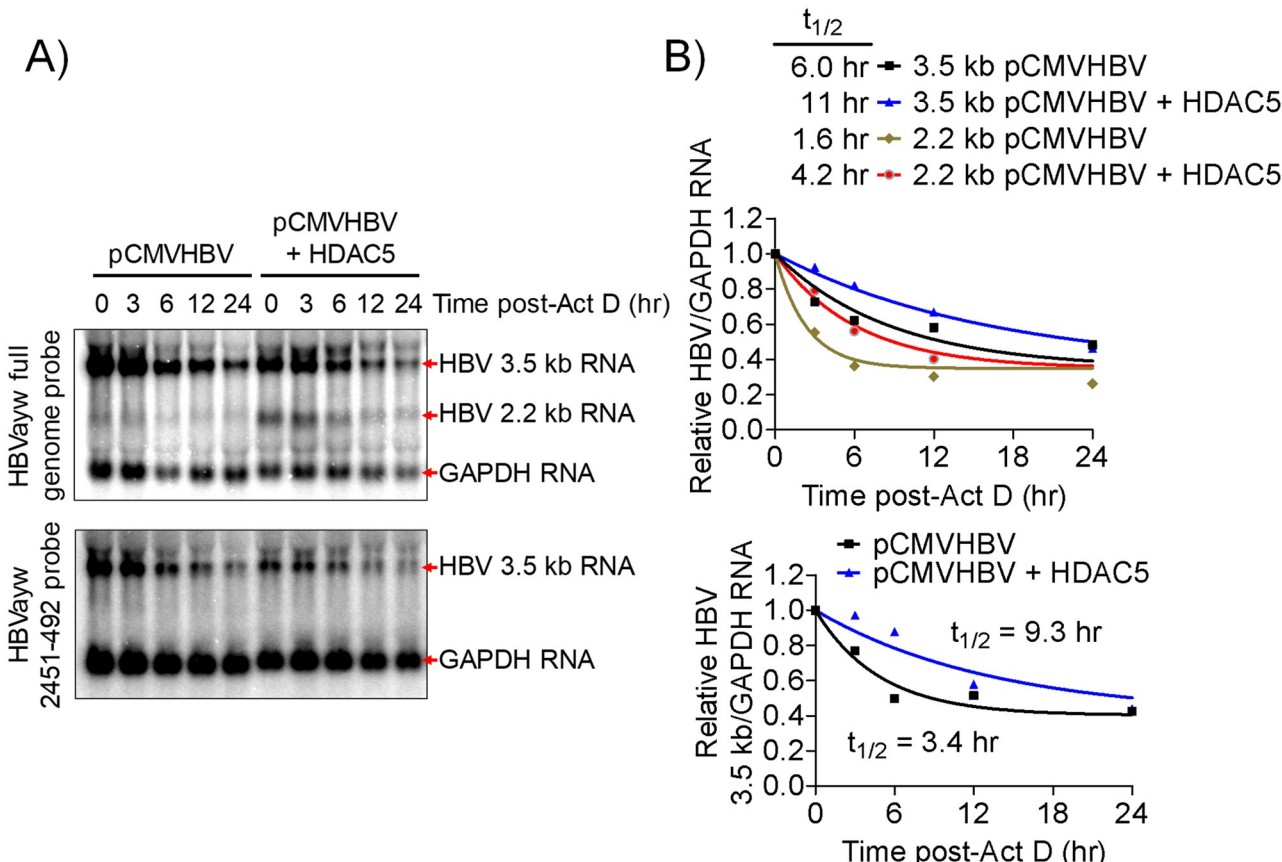

**Fig 7. HDAC5 increases the stability of the HBV RNAs.** HEK293T cells were transfected with the pCMVHBV construct with or without the HDAC5 expression vector for 40 hours. The cells were then treated with actinomycin D for up to 24 hours and total RNA collected at the indicated times. (A) RNA (Northern) filter hybridization analysis using the complete genomic HBVayw DNA probe or a subgenomic HBVayw DNA probe spanning nucleotide coordinates 2451–492. (B) Quantitation of HBV 3.5 and spliced 2.2 kb RNAs relative to GAPDH RNA. The estimated relative abundance of the HBV RNA at various time points was compared to the level when actinomycin D treatment was initiated (t = 0 hr, relative abundance of HBV RNA = 1.0). The half-life of the RNAs was determined using one phase decay analysis (GraphPad Prism 8.4).

the greater abundance of the HBV 3.5 and spliced 2.2 kb RNAs in the presence of HDAC5 appears to result from their enhanced stability, resulting in an overall increase in HBV biosynthesis.

Collectively, these results support a model for the regulation of HBV biosynthesis involving enhanced viral RNA splicing and stability mediated by HDAC5 activity (Fig 8). Although HBV replication is generally considered to be independent of RNA splicing (Fig 8, steps 1–4), splice variants and their reverse transcribed derivatives have been observed in cell culture studies and in the sera of chronically infected patients, with the 2.2 kb splice variant being the primary variant detected [6,12,13,15,16,42,47]. Here, increased splicing of the HBV 3.5 kb RNA by HDAC5 expression was associated with greater stability of both the HBV 3.5 and spliced 2.2 kb RNAs (Figs 3–7 and 8, step 6). This observation suggests that the splicing machinery might be utilized by the virus to stabilize the viral pregenomic RNA. In addition, the generation of the HBV 2.2 kb splice variant leads to the synthesis of a truncated core protein without the terminal cysteine residue (Fig 8, step 7), which efficiently forms capsids and is fully replication competent (Fig 6, [42]). The effect of HDAC5 on RNA splicing and stability could be mediated via deacetylation of a nuclear host factor (HF) that regulates these processes (Figs 2 and 8, steps 5 and 6), such as

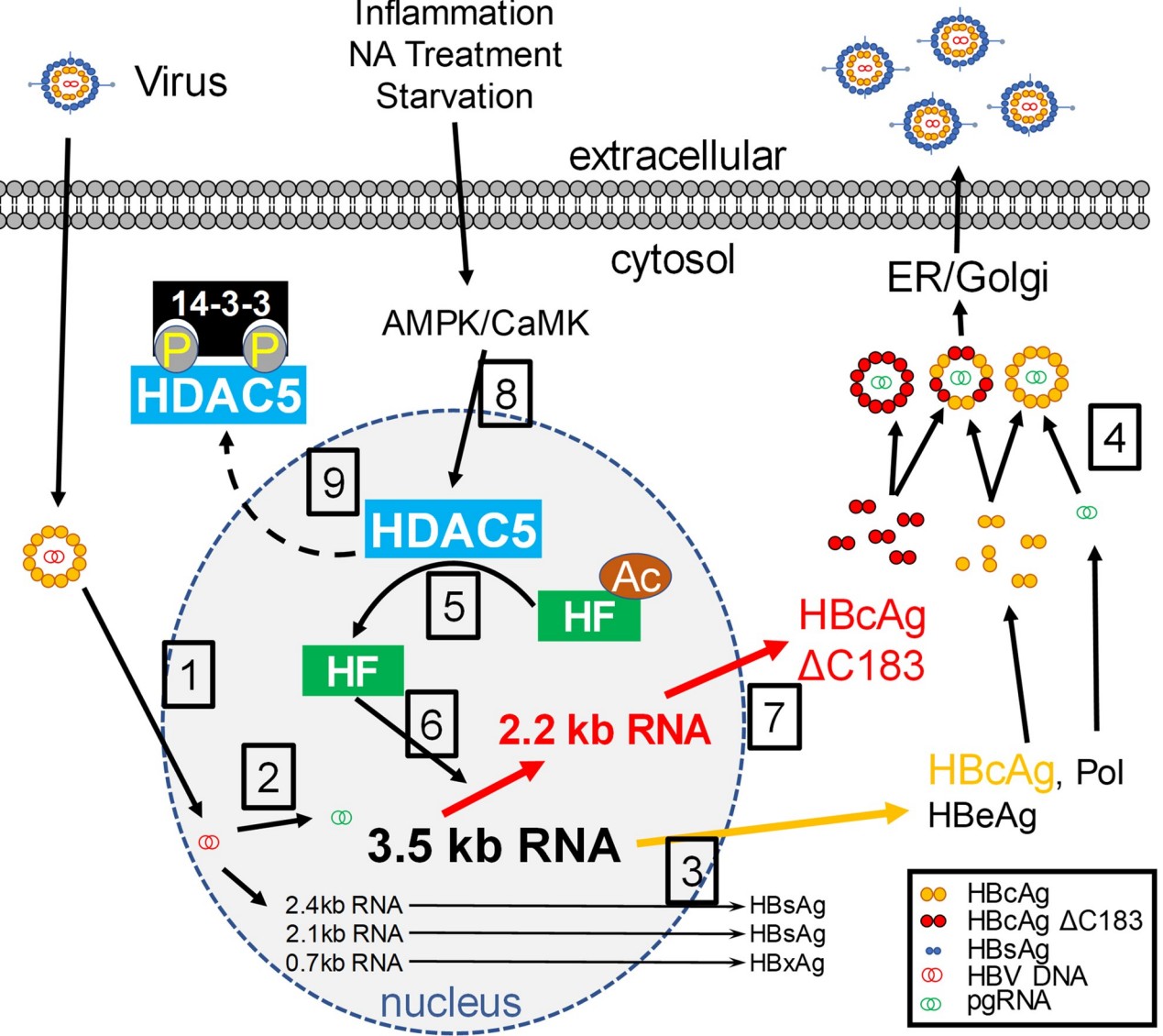

**Fig 8. Proposed model for the effect of HDAC5 on HBV biosynthesis.** (1 and 2) Virus infects hepatocytes and delivers HBV genomic DNA to the nucleus where it is converted to HBV cccDNA and transcribed to produce the viral RNAs. (3) Viral RNAs are translated to produce the viral proteins required for HBV biosynthesis. (4) Viral core protein encapsidates pregenomic RNA and polymerase protein. Reverse transcription and envelopment by budding through the ER/Golgi system ensue to produce virions. (5 and 6) HDAC5 deacetylates a host factor (HF) that is necessary, directly or indirectly, for HBV 3.5 kb RNA stability and splicing to the HBV 2.2 kb RNA. (7) HBV spliced 2.2 kb RNA is translated to terminal-cysteine deleted (ΔC183) core polypeptide, which is capable of forming replication competent capsids similar to the wild type protein in step 4 (Fig 6; [42]). (8 and 9) Multiple protein kinases phosphorylate HDAC5, leading to nuclear export and cytoplasmic sequestration of phospho-HDAC5, thereby preventing the deacetylation of nuclear HFs mediating the HBV 3.5 kb RNA stability and splicing. These kinases are responsive to environmental stimuli such as proinflammatory signals, antiviral nucleos(t)ide analogs (NA) and lack of nutrients.

an RNA binding protein and/or splicing factor [43,48]. HDAC5 binding to several splicing factors and RNA binding proteins has been shown to require its nuclear localization [48], which is regulated by a variety of kinases, including Ca$^{2+}$/calmodulin-dependent protein kinase (CaMK) [49] and 5' AMP-activated protein kinase (AMPK) [50]. These kinases phosphorylate key serine residues, leading to the cytosolic sequestration of phospho-HDAC5 by 14-3-3 proteins [49] (Fig 8, steps 8 and 9). Interestingly, activation of CaMK and AMPK has been

reported to negatively impact HBV biosynthesis, although their mechanism(s) of action are unclear [51–53]. Indeed, these kinases are activated by antiviral factors such as nucleos(t)ide analogs therapeutics [54] and proinflammatory signals, including interferon [55], poly I:C [56], lipopolysaccharide [56,57] and tumor necrosis factor alpha (TNFα) [57]. This potentially suggests a role of HDAC5 in mediating the inhibitory effects of these antiviral factors. Activation of CaMK and AMPK by these stimuli leads to the nuclear export of phopho-HDAC5 and would presumably eliminate its interaction with relevant host factors [48], resulting in reduced splicing and stability of the HBV 3.5 kb RNA and thereby down regulating viral biosynthesis. This might be reflected as a modest decrease in HBV 3.5 and 2.2 kb RNA levels, which potentially could be missed without the appropriate analysis to distinguish between the HBV spliced 2.2 kb and unspliced 2.1 kb RNAs. However, this change in viral transcript profile might result in a modest reduction in core/p21 polypeptide with an associated dramatic loss of capsid formation due to the concentration-dependent cooperative process involved in capsid assembly [35–37]. In fact, studies utilizing proinflammatory signals typically report a post-transcriptional inhibitory effect on HBV biosynthesis associated with limited changes in viral RNA levels but with a loss of cytoplasmic capsids [58,59]. In combination, these observations suggest a unique role for HDAC5 and RNA splicing in regulating HBV biosynthesis and indicate that they may represent novel host antiviral therapeutic targets.

HDACs play a pleiotropic role in cellular homeostasis with the different isoforms regulating unique cellular functions [18,21]. There have been multiple reports linking HDACs and HBV that were mainly focused on the histone deacetylation-dependent regulation of cccDNA accessibility to the transcriptional machinery [27–32]. However, comprehensive investigation of the effects of class I and II HDACs on HBV biosynthesis in hepatoma and nonhepatoma cells revealed that HDACs are not simply general repressors of HBV transcription (Fig 1). Whereas HDAC7, and to a lesser extent HDAC4, appear to repress HBV transcription, HDAC5 upregulated HBV capsid abundance and supported enhanced HBV biosynthesis in both hepatoma and nonhepatoma cells (Fig 1). Class I HDACs are regarded as the major histone deacetylase enzymes in the cell [18,21] but did not greatly modulate HBV transcription under the conditions examined in the current study (Fig 1). The absence of necessary factors to recruit class I HDACs to HBV DNA in the cell culture system used in the current study could potentially explain the lack of any major effects of class I HDAC expression on HBV biosynthesis (Fig 1) as had been suggested previously [27,28]. This possibility is supported by the finding that HBV cccDNA is decorated with epigenetic modifications associated with active transcription in model systems of HBV infection [60] and in liver tissue of chronically infected patients [61]. Together, these observations support the notion that distinct classes of HDACs modulate HBV biosynthesis in specific context-dependent manners beyond simple histone deacetylation.

The unique effect of HDAC5 on HBV biosynthesis suggests that the regulation of gene expression by HDACs is considerably more complex than the repression of transcription by the deacetylation of histones. The post-transcriptional regulation of alternative splicing and transcript stability by acetylation and deacetylation of splicing factors and/or RNA binding proteins may have global consequences for cellular RNA abundances and sequence coding content which extends beyond the regulation of HBV RNAs. As more than 90% of human genes undergo splicing [62,63], understanding of the regulation of this process by acetylation, and probably additional covalent modifications, may become increasingly important and potentially reveal an additional post-transcriptional code governing RNA function. This field of study may ultimately lead to new approaches to developing HDAC inhibitors with improved efficacy and safety profiles as both antiviral and cancer agents.

## Materials and methods

### Plasmid constructions

The HBV DNA (4.1 kbp) and pCMVHBV constructs have been reported previously [40]. The HBV DNA (4.1 kbp) mutant constructs lacking expression of HBeAg (PC-), the X gene polypeptide (X-) or both (X-PC-) have been described [64]. The pCMVHDAC1 and pCMVHDAC3-8 constructs drive the expression of C-terminal FLAG tagged HDACs 1 and 3–8 under the control of the CMV immediate early promoter (Addgene plasmids # 13819–25). The plasmid pCMVHDAC5-H1006Y was derived from plasmid pCMVHDAC5 using the Q5 site-directed mutagenesis kit (New England Biolabs) according to manufacturer's instructions. The CMV-driven HBV constructs lacking expression of specific viral proteins and/or the encapsidation sequence ε (plasmids LJ122 (S-P-), FU312 (C-), LJ145 (C-S-), EL43-1 (S-P-ε-), NU9 (C-S-ε-), and LJ196 (C-S-P-)) have been reported previously [13,65–67]. The pCMVHBcAg and pcHBc plasmids expressing the core polypeptide open reading frame have been described [68,69]. Constructs CpLuc and CpLucΔ1879–129 express the firefly luciferase gene under the control of the HBV core promoter and construct pRLCMV expresses Renilla luciferase under the control of the CMV immediate early promoter [70]. Plasmids S-P-ε-ΔC183 and S-P-ε-ΔC183(2.2) were generated from plasmid EL43-1 [65] using the Q5 site-directed mutagenesis kit (New England Biolabs) according to manufacturer's instructions. Plasmid S-P-ε-ΔC183 carries mutations T2451A and C2459A in the HBVayw sequence to replace the terminal cysteine residue with a translation stop codon. Plasmid S-P-ε-ΔC183(2.2) was generated by deleting HBVayw coordinates 2451–491.

### Cells, transfections, and biological materials

The human hepatoma HepG2 and the embryonic kidney HEK293T cell lines were grown in RPMI-1640 medium and 10% fetal bovine serum at 37°C in 5% $CO_2$. Transfections were performed as previously described [71] using 10 cm plates, containing approximately $1 \times 10^6$ cells. DNA, RNA, and protein isolations were performed 3 days post transfection. The transfected DNA mixture was composed of 10 μg of the HBV DNA (4.1 kbp) or pCMVHBV constructs plus 1–10 μg of the HDAC expression vectors plus control vectors as indicated. Controls were derived from cells transfected with HBV DNAs and the expression vector lacking a cDNA insert, pCMVPa [72]. LMK235 and actinomycin D (Sigma-Aldrich), were dissolved in DMSO and methanol, respectively. Firefly and Renilla luciferase signals were obtained using the dual-luciferase reporter assay (Promega) according to manufacturer's instructions.

### Characterization of HBV transcripts, viral replication intermediates, viral capsids, and HBcAg/p21 proteins

Transfected cells from a single plate were divided equally and used for the preparation of total cellular RNA, cytoplasmic protein, and viral DNA replication intermediates as described [37,73]. RNA (Northern) and DNA (Southern) filter hybridization analyses were performed using 10 μg of total cellular RNA and viral DNA replication intermediates derived from one third of the cells per 10 cm plate. Intracapsid HBV DNA replication intermediates were analyzed as described [74].

Transcription initiation sites for the HBV 3.5 kb transcripts were examined by primer extension analysis using 4 units of avian myeloblastosis virus reverse transcriptase (Promega), 10 ng [32]P-labeled HBV oligonucleotide probe, 5'-GGAAAGAAGTCAGAAGGCAAAAACG AGAGTAACTCC-3' (HBV nucleotide coordinates 1976 to 1941) and 10 μg of total cellular RNA as described [75]. Filter hybridization and primer extension analyses were quantified by

phosphorimaging using a Molecular Dynamics Typhoon 8600 Phosphor Imager system. Cytoplasmic lysate was prepared by lysing cells in 50 mM HEPES pH 7.5, 150 mM NaCl, 1% Igepal CA-630 (Sigma Aldrich), 5% glycerol, 1x complete protease inhibitor cocktail (Roche), and 1x phosphatase inhibitor cocktail set II (Millipore). The lysate was clarified by centrifugation at 19,000g for 10 minutes at 4˚C and protein concentration was determined using the BCA assay (Thermo Fisher). For western blot analysis of viral capsids, 50 μg of cytoplasmic lysate was loaded onto an 0.8% agarose gel in TAE and electrophoresed at 75V for 2 hours. The gel was transferred to PVDF membrane by capillary transfer in TNE buffer (10 mM Tris pH 7.5, 150 mM NaCl, and 1 mM EDTA) overnight as described previously [76]. For detection of HBcAg/p21 polypeptide, 50 μg of cytoplasmic lysate was loaded onto 4–12% Bis-Tris SDS-PAGE (Thermo Fisher) and electrophoresed for 2 hours at 100V. Proteins were transferred to nitro-cellulose membrane using the iBlot2 dry transfer system (Thermo Fisher) for 7 minutes (mode P3). Viral capsids were detected with 1:200 mouse anti-HBcAg antibody 1–5 (Santa Cruz) in 5% milk in TBST (10 mM Tris pH 7.5, 140 mM NaCl, and 0.1% v/v Tween 20). The core poly-peptide (p21) was detected with 1:100 mouse anti-HBcAg antibody 10E11 (Santa Cruz) in Odyssey blocking buffer PBS (Li-Cor). GAPDH was detected with 1:10,000 mouse antibody ab128915 (Abcam) in Odyssey blocking buffer PBS (Li-Cor). FLAG epitope tagged polypep-tides were detected with 1:1,000 mouse OctA H-5 antibody (Santa Cruz) in Odyssey blocking buffer PBS (Li-Cor). HDAC5 was detected with 1:200 mouse antibody C-11 (Santa Cruz) in 5% milk in TBST. Western blot analysis was conducted as described [76,77]. For detection of viral capsids and HDAC5, horseradish peroxidase-labeled goat anti-mouse IgG (Cell Signaling Technology, 1:5000 dilution) was used and imaged using enhanced chemiluminescence sub-strate (Thermo Fisher) and the ChemiDoc MP Imaging System (BioRad). For detection of all other proteins, anti-mouse or anti-rabbit (1:10,000 dilution) IRDye conjugated secondary anti-bodies (Li-Cor) were utilized and imaged using the Odyssey Sa imager (Li-Cor).

## Polysome profiling analysis of HBV RNAs

Polysome profiling was conducted as described previously [59,78] with modifications. HEK293T cells were grown to 30% confluence in a 10 cm plate and transfected with 3 μg pCMVHBV with or without 10 μg HDAC5 expression vector and incubated for 48 hours at 37˚C to 70–80% confluency. Cycloheximide (Sigma Aldrich) was added to 350 μM and the cells were incubated for an additional 10 minutes at 37˚C. The cells were then washed with 4 mL of PBS (154 mM NaCl, 1 mM $KH_2PO_4$, 5.6 mM $Na_2HPO4$, pH 7.4) + 350 μM cyclohexi-mide and collected in 4 mL of the same solution. The cells were lysed in 200 μL of polysome extraction buffer (250 mM KCl, 10 mM $MgCl_2$, 20 mM HEPES pH 7.5, 0.5% NP40 (Sigma Aldrich), 0.5% SDC (Sigma Aldrich), 2 mM dithiothreitol (Bio-Rad), 350 μM cycloheximide, 1x cOmplete protease inhibitor cocktail, 1x phosphatase inhibitor cocktail set II, and 5% v/v RNaseOUT (Thermo Fisher)). The cells were lysed by vortexing briefly and then were left on ice for 10 minutes with additional brief vortexing every 2–3 minutes. The lysate was clarified by centrifugation for 2 minutes at 19,000g at 4˚C and the supernatant clarified again for an additional 5 minutes. Lysate (7.5 $OD_{260}$ units) was loaded onto a preformed 10–50% sucrose gradient in lysis buffer (without protease, phosphatase, and RNase inhibitors) and centrifuged at 35,000 rpm for 3 hours at 4˚C in an SW41 Ti rotor (Beckman Coulter). The gradient was fractionated (500-μL fractions) using a piston gradient fractionator (Biocomp Instruments) coupled to UV detector (Bio-Rad). Fractions were extracted with an equal volume of phenol:chloroform:isoamyl alcohol (50:49:1) (Millipore). RNA was ethanol precipitated and analyzed by RNA (Northern) filter hybridization and primer extension analyses as described.

### RT-qPCR analysis and sequencing of PCR product

RT-qPCR analysis of HBV RNAs was conducted as described previously [34]. Semi-quantitative PCR was conducted using a 15-sec extension phase and 40 cycles. The products were run on a 6% nondenaturing acrylamide gel and visualized with ethidium bromide. The 163 bp PCR product was recovered from the gel [72] and sequenced [41].

## Supporting information

**S1 Fig. HDAC5 expression correlates with HBV biosynthesis across the liver lobule.** (A) Immunohistochemical staining of an HBV transgenic mouse liver showing the presence of nuclear HBcAg throughout the liver whereas cytoplasmic staining is located primarily in the centrolobular hepatocytes (adapted from [34]). (B) Single-cell RNAseq analysis of mouse liver tissue for HDACs and sirtuins (SIRTs) demonstrating that HDAC5 is highly expressed in the liver and exhibits zonation towards the pericentral area of the liver lobule (adapted from [33]). (C) Western blot analysis of HDAC5 expression in HEK293T and HepG2 cells. Cells were transfected with the HDAC5 expression vector, pCMVHDAC5, and whole cell lysates were analyzed for HDAC5 expression after 72 hours using the mouse anti-human HDAC5 monoclonal antibody (C-11, Santa Cruz). The level of HDAC5 expression relative to GAPDH is presented.
(TIF)

**S2 Fig. HDAC5-induced upregulation of HBV biosynthesis is mediated by HBV core protein.** (A) HepG2 cells were transfected with the pCMVHBV construct lacking expression of the polymerase and surface proteins as well as the packaging signal ε (S-P-ε-). This construct was complemented with pCMVHBV constructs that lack expression of the core polypeptide but express the polymerase polypeptide (C-S- and C-S-ε-) or the polymerase and surface polypeptides (C-), in the presence or absence of the HDAC5 expression vector. The viral proteins were analyzed from cytoplasmic or whole cell extracts by Western blot analysis for native viral capsids and core polypeptide (p21). (B) HepG2 or HEK293T cells were transfected with the HBV DNA (4.1 kbp) construct or mutant constructs that lack production of the precore (4.1 PC-), X (4.1 X-), or both proteins (4.1 X-PC-) with or without the HDAC5 expression vector. The viral proteins were collected from the cytoplasm and examined by Western blot analysis for native viral capsids.
(TIF)

**S3 Fig. HDAC5 expression does not alter transcription from the HBV core promoter.** HepG2 and HEK293T cells were transfected with the HBV core promoter firefly luciferase reporter gene (CpLuc and CpLucΔ1879–129) and the CMV-driven Renilla luciferase reporter gene (pRLCMV) constructs [70]. HBV core promoter activity was normalized to Renilla luciferase activity and reported relative to the signal obtained in the cells transfected with 0.3 μg of the CpLuc construct and 0.1 μg of the pRLCMV construct which is designated as a relative activity of 1.0.
(TIF)

**S1 Data. Excel file containing, in separate sheets, the underlying numerical data for figure panels 1A, 1B, 2A, 2B, 3B, 4A, 4D, 5B, 6, 7B, S1B, S1C and S3.**
(XLSX)

## Acknowledgments

We are grateful to Dr. Eric Verdin (The Buck Institute, Novato, CA) for the plasmids pCMVHDAC1 and pCMVHDAC3-8 (Addgene plasmids # 13819–25), Dr. Ju-Tao Guo (Baruch S. Blumberg Institute, Doylestown, PA) for plasmid pCMVHBcAg, and Dr. Haitao Guo (UPMC Hillman Cancer Center, Pittsburgh, PA) for plasmid pcHBc. We also are indebted to Dr. Alexander S. Mankin, Dr. Nora Vázquez-Laslop, Dr. Laura Sanchez, Dr. Adam Oberstein, Kyle Mangano, and Aseel Y. Taha for advice and technical assistance.

## Author Contributions

**Conceptualization:** Taha Y. Taha, Daniel D. Loeb, Pavel A. Petukhov, Alan McLachlan.

**Formal analysis:** Taha Y. Taha.

**Funding acquisition:** Alan McLachlan.

**Investigation:** Taha Y. Taha, Varada Anirudhan, Umaporn Limothai.

**Methodology:** Taha Y. Taha.

**Project administration:** Pavel A. Petukhov, Alan McLachlan.

**Resources:** Daniel D. Loeb, Alan McLachlan.

**Supervision:** Pavel A. Petukhov, Alan McLachlan.

**Validation:** Taha Y. Taha, Pavel A. Petukhov, Alan McLachlan.

**Visualization:** Taha Y. Taha, Pavel A. Petukhov, Alan McLachlan.

**Writing – original draft:** Taha Y. Taha.

**Writing – review & editing:** Taha Y. Taha, Pavel A. Petukhov, Alan McLachlan.

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
