## [Decision Letter · Decision Letter 0]

20 May 2020

Dear Dr. McLachlan,

Thank you very much for submitting your manuscript "Modulation of hepatitis B virus pregenomic RNA stability and splicing by histone deacetylase 5 enhances viral biosynthesis" for consideration at PLOS Pathogens. As with all papers reviewed by the journal, your manuscript was reviewed by members of the editorial board and by several independent reviewers. The reviewers considered that the study is novel and interesting, but they also raised substantial concerns on some of the experimental design and data interpretation. In light of the reviews (below this email), we would like to invite the resubmission of a significantly-revised version that takes into account the reviewers' comments.

We cannot make any decision about publication until we have seen the revised manuscript and your response to the reviewers' comments. Your revised manuscript is also likely to be sent to reviewers for further evaluation.

Sincerely,

Haitao Guo

Guest Editor

PLOS Pathogens

Guangxiang Luo

Section Editor

PLOS Pathogens

Kasturi Haldar

Editor-in-Chief

PLOS Pathogens

orcid.org/0000-0001-5065-158X

Michael Malim

Editor-in-Chief

PLOS Pathogens

orcid.org/0000-0002-7699-2064

Reviewer's Responses to Questions

**Part I - Summary**

Reviewer #1: This manuscript studied the impact of HDAC5 overexpression on HBV RNA, replicative DNA, core protein and particles, mostly in the HEK293T human kidney cell line transiently transfected with 4.1-kb HBV DNA construct (1.3mer?) or pCMVHBV (1.1mer construct). Considering that the phenotype from HepG2 cells (Fig. 1A) was not so impressive, and that no work in Huh7 cells was described, the physiological relevance of the findings from HEK293T cells is somewhat questionable. Moreover, since HEK293T is not a good system to recapitulate late steps of HBV lifecycle such as genome replication, the HBV parameters observed could be somewhat different from well established Huh7 and HepG2 cells due to missing liver specific co-factors.

Reviewer #2: In this manuscript, Taha and colleagues examine the association between HDAC5 and HBV RNA stability and splicing. The authors show that HDAC5 increases the stability of the HBV 3.5 kb RNA and promotes the splicing of this RNA to the 2.2 kb form, which leads to increased capsid formation. Overall, the results are clear and convincing, and the findings are potentially important as they provide new insight into both HDAC5 cellular functions and HBV replication. However, some weaknesses limit the significance of the findings.

Reviewer #3: The manuscript “MODULATION OF HEPATITIS B VIRUS PREGENOMIC RNA STABILITY AND SPLICING BY HISTONE DEACETYLASE 5 ENHANCES VIRAL BIOSYNTHESIS” by Taha et al., assesses the role of HDAC5 in HBV replication. Overall, it is a very interesting and novel finding that extends out understand of HBV gene regulation and possible mechanisms of immune modulation of HBV. The study is well designed, and data is clearly presented.

**Part II – Major Issues: Key Experiments Required for Acceptance**

Reviewer #1: 1. The most striking finding from Fig. 1 is the discordance between core protein by Western blot vs. core particles by native agarose gel, in both HepG2 cells and 293T cells (according to P8, HDAC5 increased core protein by 5 fold but core particles by 50 fold in 293T cells). Such discordance was observed throughout this study, but no explanation was provided. Did the deletion of last residue from core protein dramatically enhance core particle assembly (either as a mixture with wild-type core protein or by itself)? Or did the loss of a cysteine markedly enhance binding of the core particles to the antibody used for detection? Has the authors tried a different anti-HBc antibody?

2. Many previous studies focused on the 2.2-kb spliced pgRNA, which were cited by the authors. Did the other researchers find more efficient core particle formation or discordance between core protein and core particles similar to the current work? Also, can the 2.2-kb spliced RNA be packaged into core particles and be converted to rcDNA? From the Southern blot shown in Figs. 1 and 2, HDAC5 did not alter the size of replicative DNA.

3. In Fig. 1B, top panel, the Northern blot pattern is rather complex. There are two high molecular weight bands at where 3.5-kb was marked, and between that and the marked 2.1-kb RNA there is another band. Moreover, the 0.7-kb RNA is quite strong relative to HepG2 cells. For the lowest panel in Fig. 1B, which top band was used as 3.5-kb RNA? Also, the Southern blot image showed marked increase of replicative DNA in lane 5 similar to increase in capsids, but according to quantification at the bottom, DNA replication increased by less than 6 fold whereas capsids increased by 60 fold.

4. According to P9, line 160-162, co-transfection with 10ug of HDAC5 WT or its H1006Y mutant, there is 2-fold increase in 2.1-kb RNA. Visual inspection of Fig. 2B suggested more than 2-fold increase by the H1006Y mutant. But the increase in core particles for this mutant was nearly 1000 fold according to the bottom panel! Is such a number reliable? If so why did not the authors use the H1006Y mutant to clarify the underlying mechanism in the subsequent figures?

5. According to Fig. 3C, in Northern blot there is little change in 3.5-kb RNA but marked increase in 2.2-kb RNA. According to Fig. 5B (qPCR) (text in P12, lines 233-235), HDAC5 increased 3.5-kb RNA by 3 fold, and 2.2-kb RNA by 33 fold). Such a finding could explain the increase in both core protein and capsids, but not the much dramatic increase in capsids than core protein.

6. Fig. 6 was meant to show prolonged half-lives of both 3.5-kb and 2.2-kb RNAs in the presence of HDAC5. But why was the 3.5-kb RNA much less abundant when HDAC5 was present at time point “0”?

Reviewer #2: 1. A central finding of the manuscript is that HDAC5 expression both increases the stability of the 3.5 kb HBV transcript and promotes the splicing of this transcript to the 2.2 kb form. However, this seems counterintuitive. If 3.5 kb RNA splicing to 2.2 kb is increased, it would seem that the half-life of the unspliced 3.5 kb transcript might be decreased, rather than increased (Fig 6), at least in the nucleus. This might be understandable if only cytoplasmic RNA were being analyzed, but the methods state that total RNA was used, and the model in Fig 7 appears to imply different relative amounts of the transcripts in the nucleus.

2. Although HDAC5 overexpression is clearly affecting HBV RNA splicing and capsid formation, the cellular mechanism of this effect is not defined. As the authors speculate, it could be due to the direct regulation of the splicing machinery by HDAC5. Alternatively, it could be an indirect effect of HDAC5 on the expression of other cellular genes. Furthermore, the results section ends somewhat abruptly with the implication that expression of the truncated Core variant from the 2.2 kb transcript is driving increased HBV capsid assembly, but this could have been further explored. Although discussed, the lack of mechanistic details is nevertheless a weakness of the study.

3. The experiments are mostly all carried out using transformed cell lines (HepG2 and HEK293T) that likely have differences in HDAC expression, histone acetylation, and RNA splicing compared to normal cells. The mouse liver data shown in Figure S1 are exciting but also indirect. Figure S1B shows apparent differences in HDAC5 expression across liver zones by single-cell RNA-Seq. A similar single-cell analysis could be performed using HBV transgenic mice to show a direct relationship between HDAC5 expression and HBV RNA expression and splicing in the liver.

Reviewer #3: 1) Fig. 1B - While there is no doubt that HDAC5 increased HBV DNA RI, the quantification provided seems inconsistent with the magnitude of change in the figure. The levels are clearly much more increased than the 3- and 6-fold indicated. I would ask the authors to check this and perhaps consider assessing the DNA levels more quantitatively via qPCR.

2) The use of the term epigenetics does not seem appropriate to the effect being reported here. My albeit limited understanding is that epigenetic changes are by definition inheritable. Thus, it does not appear that this specific post-translational modification would fall under the category of an epigenetic change.

**Part III – Minor Issues: Editorial and Data Presentation Modifications**

Reviewer #1: (No Response)

Reviewer #2: 1. The HDAC5 inhibitor LMK235 only had an effect at the highest concentration (250 nM), and it is not clear that the drug is specific for HDAC5 at this concentration (Fig 2A). Biochemical analysis of specificity or confirmation by RNAi knockdown is necessary to interpret this result. Similarly, Fig 2B would be strengthened by a demonstration of increased HDAC5 activity in the HDAC5 mutant-transfected cells.

2. Steps 8 and 9 in the model (Figure 7) are not directly supported by any data in the manuscript and should be removed.

Reviewer #3: 1) Lines 163-164 – The authors state the data showing an enhanced effect with the HDAC5 mutation with enhanced enzymatic activity “indicate that HDAC5-mediated support of HBV biosynthesis requires the enzymatic activity of HDAC5”. However, technically this mutant does not show the enzymatic activity is REQUIRED (for that you would need an enzymatically dead mutation), it rather simply shows a correlation with the enzymatic activity. I would recommend a slight re-phasing to avoid overstatement.

2) The implications of this work seem rather far reaching in several areas which could be discussed to a greater extent in the discussion. Obviously, this is not a requirement for publication but a suggestion to the authors. For example:

- There is quite an extensive literature on immune mediate post-transcriptional regulation of HBV. Could this be a major mechanism behind these observations? If so, such a link to the pre-existing literature would be useful to discuss.

- How much of the “2.1kb HBV RNA” observed on NB in the literature is actually this 2.2kb spliced core transcript and how might that have impacted led to the misinterpretation of HBV NBs over the years when changes in the NB band were presumed to reflect the level of the 2.1kb HBs transcript

PLOS authors have the option to publish the peer review history of their article (what does this mean?). If published, this will include your full peer review and any attached files.

Reviewer #1: No

Reviewer #2: No

Reviewer #3: No
---

## [Decision Letter · Decision Letter 1]

6 Jul 2020

Dear Alan,

Thank you very much for submitting your manuscript "Modulation of hepatitis B virus pregenomic RNA stability and splicing by histone deacetylase 5 enhances viral biosynthesis" for consideration at PLOS Pathogens. As with all papers reviewed by the journal, your manuscript was reviewed by members of the editorial board and by several independent reviewers. The reviewers appreciated the attention to an important topic. Based on the reviews, we will consider your manuscript for publication, if you could further address the minor comments raised by  reviewer 1.

Please address the additional minor points raised by Reviewer 1. Reviewer 2 has a general comment on signal quantification of Southern/northern blots, but such issue is not limited to this specific manuscript, therefore, it is up to the authors whether they want to address it or not.

Sincerely,

Haitao Guo

Guest Editor

PLOS Pathogens

Guangxiang Luo

Section Editor

PLOS Pathogens

Kasturi Haldar

Editor-in-Chief

PLOS Pathogens

orcid.org/0000-0001-5065-158X

Michael Malim

Editor-in-Chief

PLOS Pathogens

orcid.org/0000-0002-7699-2064

Please address the additional minor points raised by Reviewer 1. Reviewer 2 has a general comment on signal quantification of Southern/northern blots, but such issue is not limited to this specific manuscript, therefore, it is up to the authors whether they want to address it or not.

Reviewer Comments (if any, and for reference):

Reviewer's Responses to Questions

**Part I - Summary**

Reviewer #1: Dr. McLachlan answered questions raised in the previous round of review. A critical question is the relevance of studies based on HEK293T cells on regulation of HBV transcription in the liver. Dr. McLachlan responded that the “reconstitution” approach in non-liver cell lines has been used by his group to successfully identify transcription factors critical for regulation in the liver. As for HDAC5, the focus of this manuscript, he presented new data (Fig. S1) showing that in HBV transgenic mice, cytoplasmic HBcAg staining is primarily in centrolobular hepatocytes correlating with high expression of HDAC5 but not other histone deacetylases. This is very interesting and would be even more provocative if similar findings can be made in infected human livers in the future. He cited three references indicating that HBV core particle assembly is cooperative such that a small increase in the level of core protein can lead to dramatic increase in core particles. He discussed spliced 2.2-kb RNA as the source of core protein vs. core particle, and suggested its limited contribution towards progeny viral DNA because of the cis-preference of the P protein. He also suggested possible sources for the extra RNA bands from Northern blot in HEK293T cells transfected with HBV DNA. The answers are satisfactory.

Reviewer #2: (No Response)

Reviewer #3: The authors have responded to all the reviewer comments to a reasonable extent

**Part II – Major Issues: Key Experiments Required for Acceptance**

Reviewer #1: none.

Reviewer #2: (No Response)

Reviewer #3: I still feel there are issues with the quantification of the blots, but this is a well known caveat of densitometry analysis. Importantly, the blots are shown, allowing readers to judge the data for themselves.

**Part III – Minor Issues: Editorial and Data Presentation Modifications**

Reviewer #1: 1. Fig. S1 showed that exogenous HDAC5 was poorly expressed in HepG2 cells relative to HEK293 cells, which could explain its weaker effect when transfected to HepG2 cells. My question is whether an anti-HDAC5 antibody (thus capable of detecting endogenous protein) or anti-FLAG antibody (only recognizing exogenous HDAC5) was used for Western blot? It is an interesting question as to why (exogenous) HDAC5 is not expressed to high level in HepG2 cells, and the extent with which endogenous HDAC5 contributes to HBV transcriptional control in this particular liver cell line. This issue warrants further investigation in the future. In my opinion silencing/knockout experiments in hepatoma cell lines will complement the overexpression approach in non-liver cell lines such as HEK293T.

2. In the new Fig. 6, deltaC183 (core protein lacking the terminal residue of cysteine) was more efficient at core particle assembly, especially when mixed with full-length (183-aa) core protein. This is an interesting observation which could partly explain a more dramatic increase in core particles than core protein by HDAC5, and might be a driving force for the selection of the 2.2-kb spliced pgRNA in some patients. In this regard HCC patients infected with genotype C often harbor the Q182* nonsense mutation to remove the C-terminal 2 residues (QC) (Zhou et al., 2017, Virus Research 235: 86-95), and Dr. Nassal previously found that the C-terminal 10 residues of core protein are dispensable for genome replication (Nassal M, 1992, J Virol 66: 4107-4116). Whether the two-aa deletion in core protein also facilitates core particle assembly just like the one-aa deletion is an interesting topic for future investigation.

Reviewer #2: (No Response)

Reviewer #3: (No Response)

PLOS authors have the option to publish the peer review history of their article (what does this mean?). If published, this will include your full peer review and any attached files.

Reviewer #1: No

Reviewer #2: No

Reviewer #3: No
---

## [Editor Report · Decision Letter 2]

13 Jul 2020

Dear Dr. McLachlan,

We are pleased to inform you that your manuscript 'Modulation of hepatitis B virus pregenomic RNA stability and splicing by histone deacetylase 5 enhances viral biosynthesis' has been provisionally accepted for publication in PLOS Pathogens.

Best regards,

Haitao Guo

Guest Editor

PLOS Pathogens

Guangxiang Luo

Section Editor

PLOS Pathogens

Kasturi Haldar

Editor-in-Chief

PLOS Pathogens

orcid.org/0000-0001-5065-158X

Michael Malim

Editor-in-Chief

PLOS Pathogens

orcid.org/0000-0002-7699-2064